# CORL: Research-oriented Deep Offline Reinforcement Learning Library

**Denis Tarasov**
Tinkoff
den.tarasov@tinkoff.ai

**Alexander Nikulin**
Tinkoff
a.p.nikulin@tinkoff.ai

**Dmitry Akimov**
Tinkoff
d.akimov@tinkoff.ai

**Vladislav Kurenkov**
Tinkoff
v.kurenkov@tinkoff.ai

**Sergey Kolesnikov**
Tinkoff
s.s.kolesnikov@tinkoff.ai

## Abstract

CORL[1] is an open-source library that provides single-file implementations of Deep Offline Reinforcement Learning algorithms. It emphasizes a simple developing experience with a straightforward codebase and a modern analysis tracking tool. In CORL, we isolate methods implementation into distinct single files, making performance-relevant details easier to recognise. Additionally, an experiment tracking feature is available to help log metrics, hyperparameters, dependencies, and more to the cloud. Finally, we have ensured the reliability of the implementations by benchmarking a commonly employed D4RL benchmark.

## 1 Introduction

Deep Offline Reinforcement Learning (Deep ORL) [24] has been showing significant advancements in numerous domains such as robotics [30, 21], autonomous driving [6] and recommender systems [4]. Due to such rapid development, many open-source ORL solutions[2] emerged to help RL practitioners understand and improve well-known ORL techniques in different fields. On the one hand, they introduce ORL algorithms standard interfaces and user-friendly APIs, simplifying ORL methods incorporation into *existing* projects. On the other hand, introduced abstractions may hinder the learning curve for newcomers and the ease of adoption for researchers interested in developing *new* algorithms. One needs to understand the modularity design (several files on average), which (1) can be comprised of thousands of lines of code or (2) can hardly fit for a novel method[3].

In this technical report, we take a different perspective on an ORL library. We propose CORL (Clean Offline Reinforcement Learning) – minimalistic and isolated single-file implementations of deep ORL algorithms, that are backed up by open-sourced D4RL benchmark results. The unadorned design allows practitioners to read and understand the implementations of the algorithms straightforwardly. Moreover, CORL supports optional integration with experiments tracking tool such as Weighs&Biases[4]. This provides practitioners with a convenient way to analyze the results and behavior of all algorithms, not merely relying on a final performance commonly reported in papers.

We hope that CORL library will help ORL newcomers to study implemented algorithms and aid the researchers in quickly modifying existing methods without a need to fight through different levels of

---

[1]CORL Repository: `https://github.com/tinkoff-ai/CORL`
[2]`https://github.com/hanjuku-kaso/awesome-offline-rl#oss`
[3]`https://github.com/takuseno/d3rlpy/issues/141`
[4]wandb.ai/

3rd Offline Reinforcement Learning Workshop at Neural Information Processing Systems, 2022.

abstraction. Finally, the obtained results may serve as a point of reference for D4RL benchmarks without a need to re-implement existing algorithms and tune hyperparameters.

Yaml Configuration File          Single-File Implementation          Experiment Tracking Log

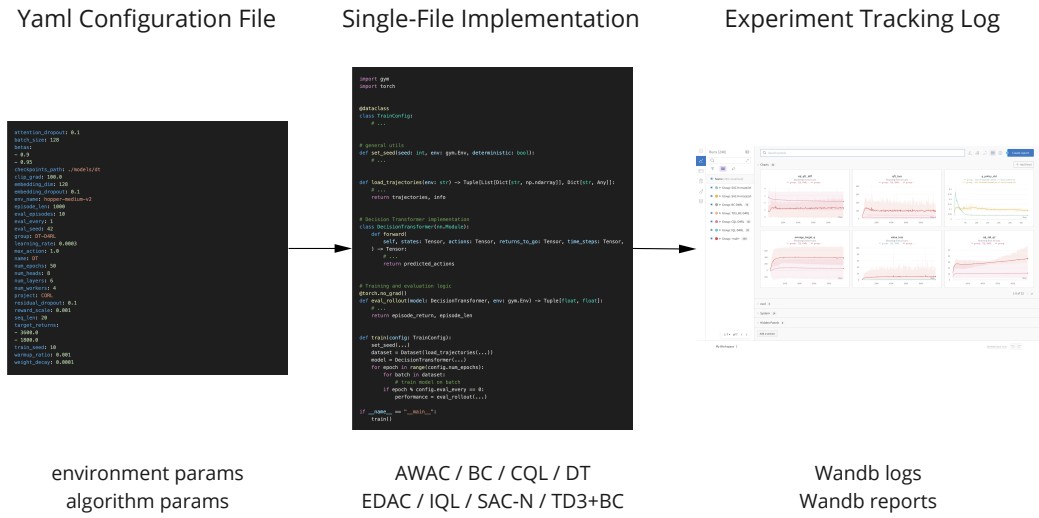

environment params                    AWAC / BC / CQL / DT                    Wandb logs
algorithm params                  EDAC / IQL / SAC-N / TD3+BC                Wandb reports

python dt.py --config=cfg/dt-hopper.yaml --logdir=logs/dt-hopper --num-epochs=50

Figure 1: The illustration of the CORL library design. Single-file implementation takes a yaml configuration file with both environment and algorithm parameters to run the experiment, which logs all required statistics to wandb.

## 2    Related Work

Since the Atari breakthrough [28], numerous open-source RL frameworks and libraries have been developed over the last years: [5, 15, 2, 13, 17, 12, 7, 19, 11, 25, 11, 26, 16, 33, 31], focusing on different perspectives of the RL. For example, stable-baselines ([15]) provides many deep RL implementations that carefully reproduce results to back up RL practitioners with reliable baselines during methods comparison. On the other hand, Ray ([25]) is focusing on implementations scalability and production-friendly usage. Finally, more nuanced solutions exist, such as Dopamine ([2]), which emphasizes different DQN variants, or ReAgent ([13]), which applies RL for the RecSys domain.

At the same time, the ORL branch, which we are interested in this paper, is not yet covered as much: the only library that precisely focus on offline RL setting is d3rlpy [32]. While CORL do also focus on ORL methods, similar to d3rlpy, it takes a different perspective on library design and provides *non-modular* independent algorithms implementations. More precisely, CORL does not introduce additional abstractions to make ORL more general but instead gives an "easy-to-hack" starter kit for research needs.

Although CORL does not represent a first non-modular RL library, which is more likely the CleanRL [16] case, it has two significant differences with its predecessor. First, CORL is focused on *offline* RL, while CleanRL implements *online* RL algorithms. Second, CORL intent to minimize the complexity of the requirements and external dependencies. To be more concrete, CORL does not have additional requirements with useful abstractions such as $stable\text{-}baselines$ or $envpool$ but instead implements everything from scratch in the codebase.

Table 1: Normalized performance of the last trained policy on D4RL averaged over 4 random seeds.

| Task Name | BC | BC-10% | TD3+BC | CQL | IQL | AWAC | SAC-$N$ | EDAC | DT |
|---|---|---|---|---|---|---|---|---|---|
| halfcheetah-medium-v2 | 42.40±0.21 | 42.46±0.81 | 48.10±0.21 | 47.08±0.19 | 48.31±0.11 | 50.01±0.30 | 68.20±1.48 | 67.70±1.20 | 42.20±0.30 |
| halfcheetah-medium-expert-v2 | 55.95±8.49 | 90.10±2.83 | 90.78±6.98 | 95.98±0.83 | 94.55±0.21 | 95.29±0.91 | 98.96±10.74 | 104.76±0.74 | 91.55±1.10 |
| halfcheetah-medium-replay-v2 | 35.66±2.68 | 23.59±8.02 | 44.84±0.68 | 45.19±0.58 | 43.53±0.43 | 44.91±1.30 | 60.70±1.17 | 62.06±1.27 | 38.91±0.57 |
| hopper-medium-v2 | 53.51±2.03 | 55.48±8.43 | 60.37±4.03 | 64.98±6.12 | 62.75±6.02 | 63.69±4.29 | 40.82±11.44 | 101.70±0.32 | 65.10±1.86 |
| hopper-medium-expert-v2 | 52.30±4.63 | 111.16±1.19 | 101.17±10.48 | 93.89±14.34 | 106.24±6.09 | 105.29±7.19 | 101.31±13.43 | 105.19±11.64 | 110.44±0.39 |
| hopper-medium-replay-v2 | 29.81±2.39 | 70.42±9.99 | 64.42±24.84 | 87.67±14.42 | 84.57±13.49 | 98.15±2.85 | 100.33±0.90 | 99.66±0.94 | 81.77±7.93 |
| walker2d-medium-v2 | 63.23±18.76 | 67.34±5.97 | 82.71±5.51 | 80.38±3.45 | 84.03±5.42 | 69.39±31.97 | 87.47±0.76 | 93.36±1.60 | 67.63±2.93 |
| walker2d-medium-expert-v2 | 98.96±18.45 | 108.70±0.29 | 110.03±0.41 | 109.68±0.52 | 111.68±0.56 | 111.16±2.41 | 114.93±0.48 | 114.75±0.86 | 107.11±1.11 |
| walker2d-medium-replay-v2 | 21.80±11.72 | 54.35±7.32 | 85.62±4.63 | 79.24±4.97 | 82.55±8.00 | 71.73±13.98 | 78.99±0.58 | 87.10±3.21 | 59.86±3.15 |
| **locomotion avg** | 50.40 | 69.29 | 76.45 | 78.23 | 79.80 | 78.85 | 83.52 | 92.92 | 73.84 |
| maze2d-umaze-v1 | 0.36±10.03 | 12.18±4.95 | 29.41±14.22 | -14.83±0.47 | 37.69±1.99 | 68.30±25.72 | 130.59±19.08 | 95.26±7.37 | 18.08±29.35 |
| maze2d-medium-v1 | 0.79±3.76 | 14.25±2.69 | 59.45±41.86 | 86.62±11.11 | 35.45±0.98 | 82.66±46.71 | 88.61±21.62 | 57.04±3.98 | 31.71±30.40 |
| maze2d-large-v1 | 2.26±5.07 | 11.32±5.88 | 97.10±29.34 | 33.22±43.66 | 49.64±22.02 | 218.87±3.96 | 204.76±1.37 | 95.60±26.46 | 35.66±32.56 |
| **maze2d avg** | 1.13 | 12.58 | 61.99 | 35.00 | 40.92 | 123.28 | 141.32 | 82.64 | 28.48 |
| antmaze-umaze-v0 | 51.50±8.81 | 67.75±6.40 | 93.25±1.50 | 72.75±5.32 | 74.50±11.03 | 63.50±9.33 | 0.00±0.00 | 29.25±33.35 | 51.75±11.76 |
| antmaze-medium-play-v0 | 0.00±0.00 | 2.50±1.91 | 0.00±0.00 | 0.00±0.00 | 71.50±12.56 | 0.00±0.00 | 0.00±0.00 | 0.00±0.00 | 0.00±0.00 |
| antmaze-large-play-v0 | 0.00±0.00 | 0.00±0.00 | 0.00±0.00 | 0.00±0.00 | 40.75±12.69 | 0.00±0.00 | 0.00±0.00 | 0.00±0.00 | 0.00±0.00 |
| **antmaze avg** | 17.17 | 23.42 | 31.08 | 24.25 | 62.25 | 21.17 | 0.00 | 9.75 | 17.25 |
| **total avg** | 33.90 | 48.77 | 64.48 | 58.79 | 68.52 | 76.20 | 78.38 | 74.23 | 53.45 |

## 3   CORL Design

**Single-File Implementations**

It is known that implementation subtleties significantly impact agent performance in deep RL [14, 8, 10]. Unfortunately, user-friendly abstractions and general interfaces, the core idea behind modular libraries, encapsulate and often hide these important nuances from the practitioners. For such a reason, CORL unwraps these details by adopting single-file implementations. To be more concrete, we put environment details, algorithms hyperparameters, and evaluation parameters into a single file [5]. For example, we have a

- $any\_percent\_bc.py$ (399 LOC[6]) as a baseline algorithm for ORL methods comparison,
- $td3\_bc.py$ (507 LOC) as a competitive minimalistic ORL algorithm [10],
- $dt.py$ (542 LOC) as an example of the recently proposed trajectory optimization approach [3]

Figure 1 depicts an overall library design. While such design produces code duplications among implementations, it has several essential benefits from the both educational and research perspective:

- **Smooth learning curve**. Having the entire code in one place makes understanding all its aspects more straightforward. In other words, one may find it easier to dive into 512 LOC of single-file Decision Transformer implementation rather than 10+ files of the original implementation.

- **Simple prototyping**. As we are not interested in code general applicability, we could make it implementation-specific. Such a design also removes the need for inheritance from general primitives or their refactoring, reducing abstraction overhead to zero. At the same time, this idea gives us complete freedom during code modification.

- **Faster debugging**. Without additional abstractions, implementation simplifies to a single for-loop with a global python name scope. Furthermore, such flat architecture makes it easier to access and inspect any created variable during the training process, which is crucial during modification and debugging.

**Configuration files**

Although it is a typical pattern to use a command line interface (CLI) for single-file experiments in the research community, CORL slightly improves it with predefined configuration files. Utilizing YAML parsing through CLI, for each experiment, we gather all environment and algorithm hyperparameters

---

[5]We follow the PEP8 style guide with a maximum line length of 89, which increases LOC a bit.
[6]Lines Of Code

Table 2: Normalized performance of the best trained policy on D4RL averaged over 4 random seeds.

| Task Name | BC | BC-10% | TD3+BC | CQL | IQL | AWAC | SAC-$N$ | EDAC | DT |
|---|---|---|---|---|---|---|---|---|---|
| halfcheetah-medium-v2 | 43.60±0.16 | 43.90±0.15 | 48.93±0.13 | 47.45±0.10 | 48.77±0.06 | 50.87±0.21 | 72.21±0.35 | 69.72±1.06 | 42.73±0.11 |
| halfcheetah-medium-expert-v2 | 79.69±3.58 | 94.11±0.25 | 96.59±1.01 | 96.74±0.14 | 95.83±0.38 | 96.87±0.31 | 111.73±0.55 | 110.62±1.20 | 93.40±0.25 |
| halfcheetah-medium-replay-v2 | 40.52±0.22 | 42.27±0.53 | 45.84±0.30 | 46.38±0.14 | 45.06±0.16 | 46.57±0.27 | 67.29±0.39 | 66.55±1.21 | 40.31±0.32 |
| hopper-medium-v2 | 69.04±3.35 | 73.84±0.43 | 70.44±1.37 | 77.47±6.00 | 80.74±1.27 | 99.40±1.12 | 101.79±0.23 | 103.26±0.16 | 69.42±4.21 |
| hopper-medium-expert-v2 | 90.63±12.68 | 113.13±0.19 | 113.22±0.50 | 112.74±0.07 | 111.79±0.47 | 113.37±0.63 | 111.24±0.17 | 111.80±0.13 | 111.18±0.24 |
| hopper-medium-replay-v2 | 68.88±11.93 | 90.57±2.38 | 98.12±1.34 | 102.20±0.38 | 102.33±0.44 | 101.76±0.43 | 103.83±0.61 | 103.28±0.57 | 88.74±3.49 |
| walker2d-medium-v2 | 80.64±1.06 | 82.05±1.08 | 86.91±0.32 | 84.57±0.15 | 87.99±0.83 | 86.22±4.58 | 90.17±0.63 | 95.78±1.23 | 74.70±0.64 |
| walker2d-medium-expert-v2 | 109.95±0.72 | 109.90±0.10 | 112.21±0.07 | 111.63±0.20 | 113.19±0.33 | 113.40±2.57 | 116.93±0.49 | 116.52±0.86 | 108.71±0.39 |
| walker2d-medium-replay-v2 | 48.41±8.78 | 76.09±0.47 | 91.17±0.83 | 89.34±0.59 | 91.85±2.26 | 87.06±0.93 | 85.18±1.89 | 89.69±1.60 | 68.22±1.39 |
| **locomotion avg** | 70.15 | 80.65 | 84.83 | 85.39 | 86.40 | 88.39 | 95.60 | 96.36 | 77.49 |
| maze2d-umaze-v1 | 16.09±1.00 | 22.49±1.75 | 99.33±18.66 | 84.92±34.40 | 44.04±3.02 | 141.92±12.88 | 153.12±7.50 | 149.88±2.27 | 63.83±20.04 |
| maze2d-medium-v1 | 19.16±1.44 | 27.64±2.16 | 150.93±4.50 | 137.52±9.83 | 92.25±40.74 | 160.95±11.64 | 93.80±16.93 | 154.41±1.82 | 68.14±14.15 |
| maze2d-large-v1 | 20.75±7.69 | 41.83±4.20 | 197.64±6.07 | 153.29±12.86 | 138.70±44.70 | 228.00±2.06 | 207.51±1.11 | 182.52±3.10 | 50.25±22.33 |
| **maze2d avg** | 18.67 | 30.65 | 149.30 | 125.25 | 91.66 | 176.96 | 151.48 | 162.27 | 60.74 |
| antmaze-umaze-v0 | 71.25±9.07 | 79.50±2.38 | 97.75±1.50 | 85.00±3.56 | 87.00±2.94 | 74.75±8.77 | 0.00±0.00 | 75.00±27.51 | 60.50±3.11 |
| antmaze-medium-play-v0 | 4.75±2.22 | 8.50±3.51 | 6.00±2.00 | 3.00±0.82 | 86.00±2.16 | 14.00±11.80 | 0.00±0.00 | 0.00±0.00 | 0.25±0.50 |
| antmaze-large-play-v0 | 0.75±0.50 | 11.75±2.22 | 0.50±0.58 | 0.50±0.58 | 53.00±6.83 | 0.00±0.00 | 0.00±0.00 | 0.00±0.00 | 0.00±0.00 |
| **antmaze avg** | 25.58 | 33.25 | 34.75 | 29.50 | 75.33 | 29.58 | 0.00 | 25.00 | 20.25 |
| **total avg** | 50.94 | 61.17 | 87.71 | 82.18 | 85.24 | 94.34 | 87.65 | 95.27 | 62.69 |

into such files so that you could use them as an initial setup. We found that such setup (1) simplifies experiments, eliminating the need to keep all algorithm-environment-specific parameters in mind, and (2) keeps it convenient with the familiar CLI approach.

**Experiment Tracking**

ORL evaluation is another challenging aspect of the current ORL state [23]. To face this uncertainty, CORL supports integration with Wandb, a modern experiment tracking tool. With each experiment, CORL automatically saves: (1) source code, (2) dependencies (requirements.txt), (3) hardware setup, (4) OS environment variables, (5) hyperparameters, (6) training and system metrics, (7) logs (stdout, stderr). See Appendix B for an example.

Although, Wandb is a proprietary solution, other alternatives such as Tensorboard or Aim could be used within a few lines of code change. It is also important to note that with Wandb tracking, one could straightforwardly use CORL with Wandb hyperparameter tuning or public reports.

We found full metrics tracking during the training process necessary for two reasons. First, it removes the possible bias of final or best performance commonly reported in papers. For example, one could evaluate ORL performance as max archived score, while another uses the average performance over $N$ (last) evaluations [32]. Second, it opens an opportunity for advanced performance analysis such as EOP [23]. In short, provided with all metrics logs, one can utilize any performance statistics, not merely relying on commonly used alternatives.

## 4   Benchmarks

In our library we implemented the following algorithms: $N\%$[7] Behavioral Cloning (BC), TD3 + BC [10], CQL [22], IQL [20], AWAC [29], SAC-N, EDAC [1], and Decision Transformer (DT) [3]. We evaluated every algorithm on the D4RL benchmark [9], focusing on Locomotion, Maze2D, and AntMaze tasks. Each algorithm was run for 1 million gradient steps[8] and evaluated every 5000 [8] steps using 10 and 100 episodes for locomotion and maze tasks, respectively. For our experiments, we used hyperparameters proposed in the original works (see Appendix C for details).

The final performance results are reported in Tables 1 and 2. The scores are normalized to the range between 0 and 100 [9]. Following recent works [32] we report both last (Table 1) and best (Table 2) obtained scores to illustrate potential performance and overfitting properties of each algorithm. See Appendix A for full training performance graphs.

According to the observed results, AWAC, SAC-N, and EDAC show the most competitive performance in both the last and best evaluation settings. At the same time, TD3 + BC performs well across all

---

[7]$N$ is a percentage of best trajectories with the highest return used for training. We omit percentage when it is equal to 100.

[8]Except SAC-$N$, EDAC and DT due to their original hyperparameters. See Appendix C for details.

tasks without any hyperparameters tuning while other methods rely on it. Finally, simple yet effective BC-10% performs well on locomotion tasks when high-quality data is available and converges there almost instantly (see Appendix A).

## 5   Conclusion

In this paper, we introduced CORL – a single-file implementation library for Offline Reinforcement Learning with configuration files and advanced metrics tracking support. All implemented algorithms were benchmarked on D4RL datasets, closely matching (sometimes overperforming) the original results. Focusing on implementation clarity and reproducibility, we hope that CORL will help RL practitioners in their research and applications.

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

# A    Additional Benchmark Information

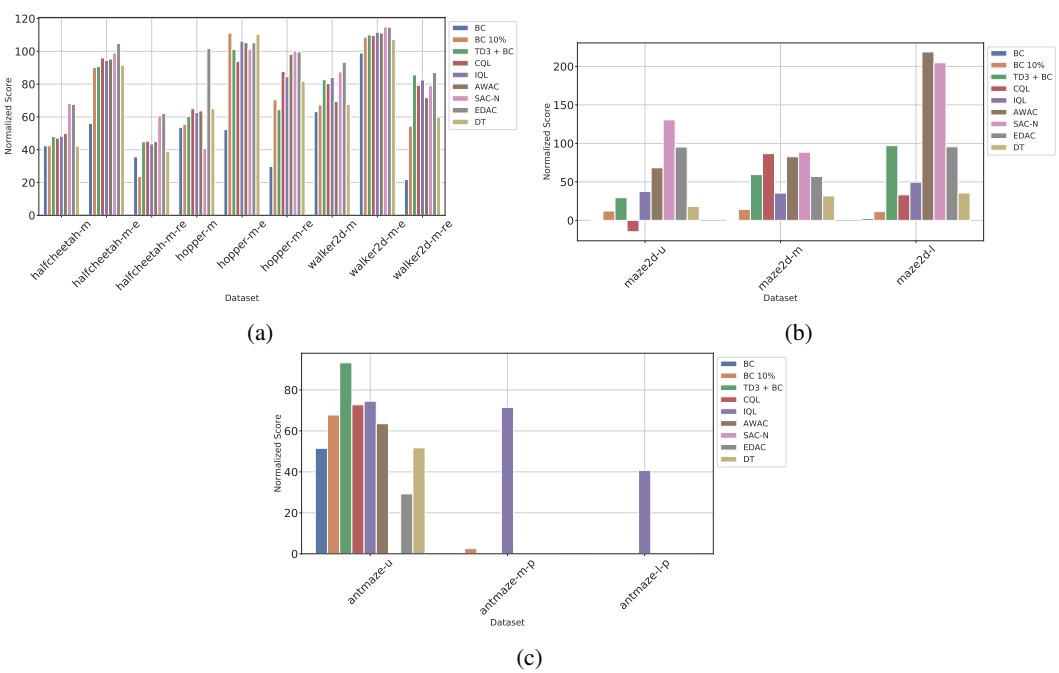

(a)

(b)

(c)

Figure 2: Graphical representation of the normalized performance of the last trained policy on D4RL averaged over 4 random seeds. (a) Locomotion datasets. (b) Maze2d datasets (c) AntMaze datasets

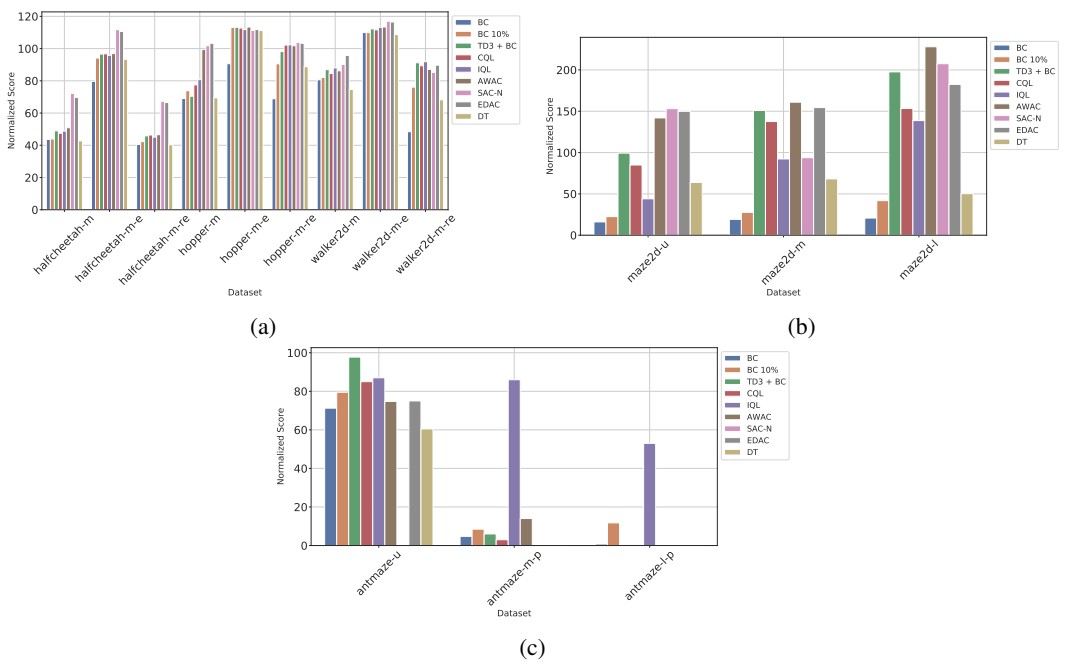

(a)

(b)

(c)

Figure 3: Graphical representation of the normalized performance of the best trained policy on D4RL averaged over 4 random seeds. (a) Locomotion datasets. (b) Maze2d datasets (c) AntMaze datasets

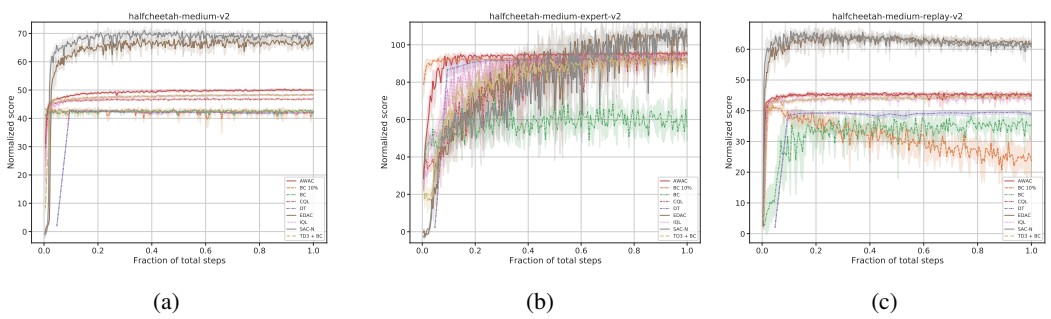

(a)                 (b)                 (c)

Figure 4: Training curves for HalfCheetah task.
(a) Medium dataset, (b) Medium-expert dataset, (c) Medium-replay dataset

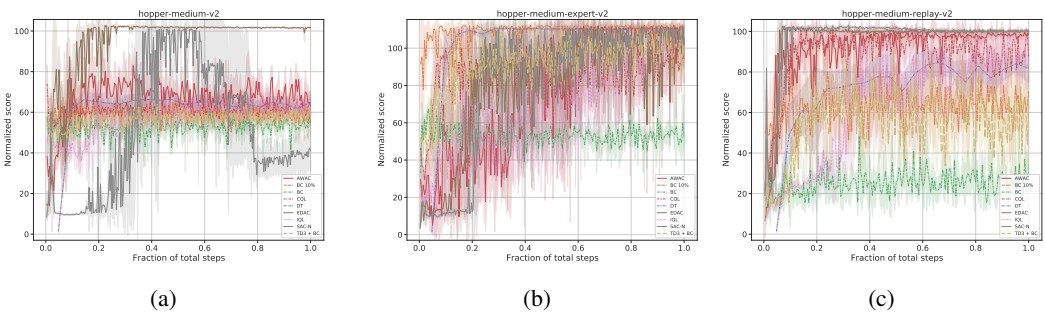

(a)                 (b)                 (c)

Figure 5: Training curves for Hopper task.
(a) Medium dataset, (b) Medium-expert dataset, (c) Medium-replay dataset

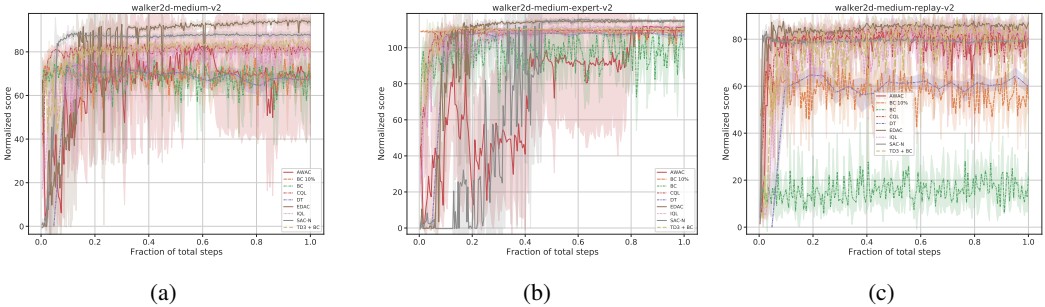

(a)  (b)  (c)

Figure 6: Training curves for Walker2d task.
(a) Medium dataset, (b) Medium-expert dataset, (c) Medium-replay dataset

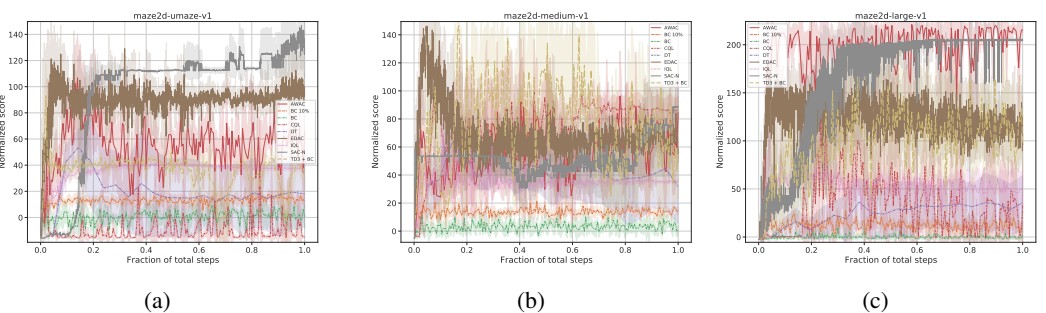

(a)  (b)  (c)

Figure 7: Training curves for Maze2d task.
(a) Medium dataset, (b) Medium-expert dataset, (c) Medium-replay dataset

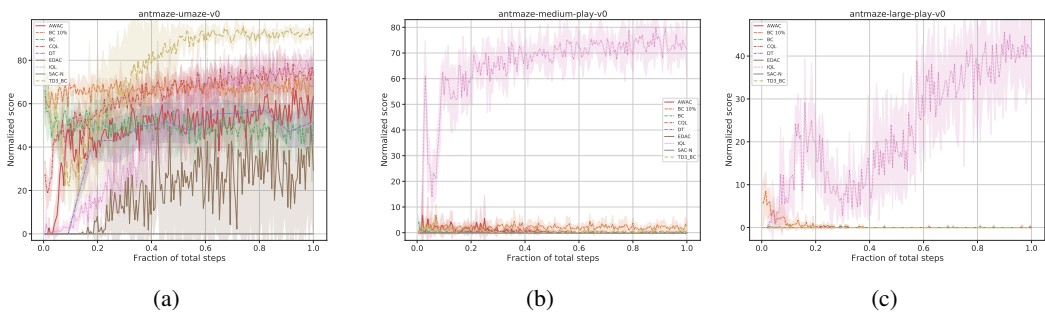

(a)  (b)  (c)

Figure 8: Training curves for AntMaze task.
(a) Umaze dataset, (b) Medium-play dataset, (c) Large-play dataset

# B Wandb Tracking

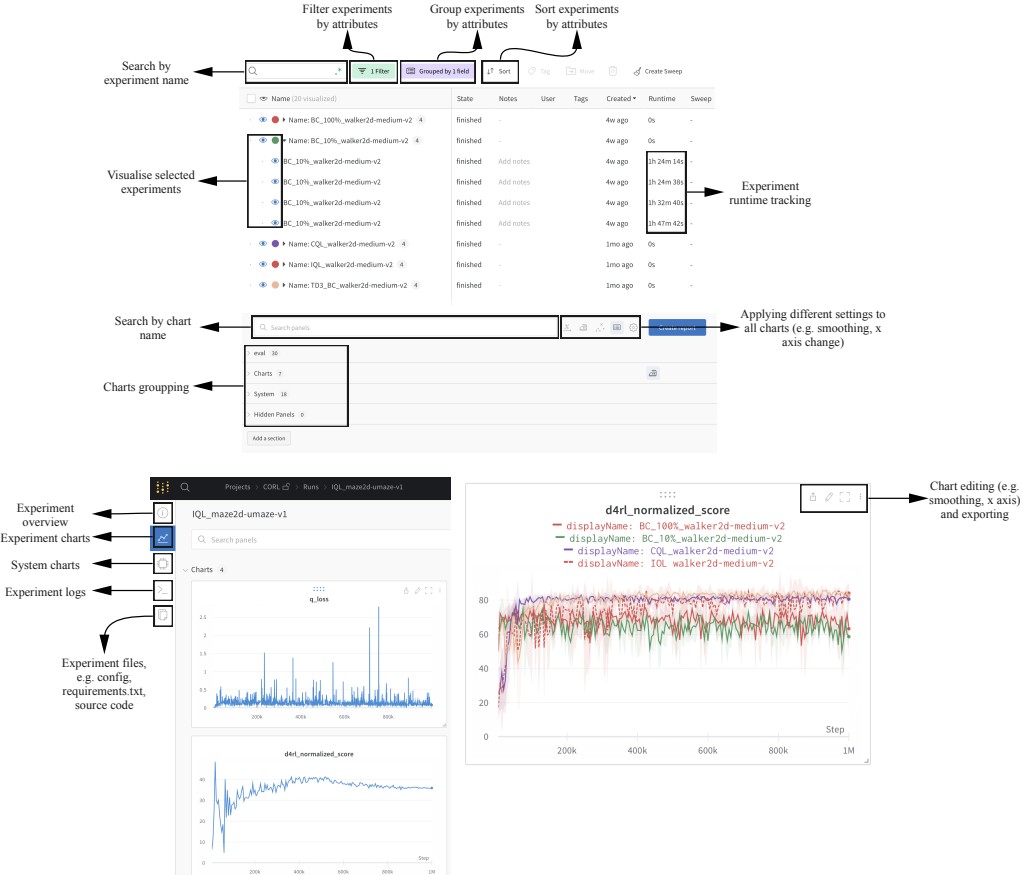

Figure 9: Screenshots of Wandb experiment tracking interface.

# C   Experimental Details

We modify reward on AntMaze task by substructing 1 from reward as it is done in previous works.

We used original implementation of TD3 + BC[9], SAC-$N$[10] and EDAC[10] and custom implementations of IQL[11] and CQL[12] as the basis for ours. For most of the algorithms and datasets we use default hyperparameters if available. Configuration files for every algorithm and environment are presented in our GitHub repository. Hyperparameters are also provided in subsection C.2.

## C.1   Number of update steps and evaluation rate

Following original work SAC-$N$ and EDAC are trained for 3 million steps (except AntMaze which is trained for 1 million steps) in order to obtain state-of-the-art performance and tested every 10000 steps. Decision Transformer (DT) training is splitted into datasets pass epochs, we train DT for 50 epochs on each dataset and evaluate every 5 epochs. All other algorithms are trained for 1 million steps and evaluated every 5000 steps. We evaluate every policy for 10 episodes on locomotion tasks and for 100 for Maze2d and AntMaze tasks.

## C.2   Hyperparameters

Table 3: BC and BC-$N\%$ hyperparameters. † used for the best trajectories choice.

|  | Hyperparameter | Value |
|---|---|---|
| BC hyperparameters | Optimizer | Adam [18] |
|  | Learning Rate | 3e-4 |
|  | Mini-batch size | 256 |
| Architecture | Policy hidden dim | 256 |
|  | Policy hidden layers | 2 |
|  | Policy activation function | ReLU |
| BC-$N\%$ hyperparameters | Ratio of best trajectories used | 0.1 |
|  | Discount factor† | 1.0 |
|  | Max trajectory length† | 1000 |

---

[9]https://github.com/sfujim/TD3_BC
[10]https://github.com/snu-mllab/EDAC
[11]https://github.com/gwthomas/IQL-PyTorch
[12]https://github.com/young-geng/CQL

Table 4: TD3+BC hyperparameters.

|  | Hyperparameter | Value |
|---|---|---|
| TD3 hyperparameters | Optimizer | Adam [18] |
|  | Critic learning rate | 3e-4 |
|  | Actor learning rate | 3e-4 |
|  | Mini-batch size | 256 |
|  | Discount factor | 0.99 |
|  | Target update rate | 5e-3 |
|  | Policy noise | 0.2 |
|  | Policy noise clipping | (-0.5, 0.5) |
|  | Policy update frequency | 2 |
| Architecture | Critic hidden dim | 256 |
|  | Critic hidden layers | 2 |
|  | Critic activation function | ReLU |
|  | Actor hidden dim | 256 |
|  | Actor hidden layers | 2 |
|  | Actor activation function | ReLU |
| TD3+BC hyperparameters | $\alpha$ | 2.5 |

Table 5: CQL hyperparameters. Note: used hyperparameters are suboptimal on AntMaze for the implementation we provide.

|  | Hyperparameter | Value |
|---|---|---|
| SAC hyperparameters | Optimizer | Adam [18] |
|  | Critic learning rate | 3e-4 |
|  | Actor learning rate | 3e-5 |
|  | Mini-batch size | 256 |
|  | Discount factor | 0.99 |
|  | Target update rate | 5e-3 |
|  | Target entropy | $-1 \cdot$ Action Dim |
|  | Entropy in Q target | False |
| Architecture | Critic hidden dim | 256 |
|  | Critic hidden layers | 3 |
|  | Critic activation function | ReLU |
|  | Actor hidden dim | 256 |
|  | Actor hidden layers | 3 |
|  | Actor activation function | ReLU |
| CQL hyperparameters | Lagrange | True, Maze2d False, otherwise |
|  | $\alpha$ | 10 |
|  | Lagrange gap | 5, Maze2d |
|  | Pre-training steps | 0 |
|  | Num sampled actions (during eval) | 10 |
|  | Num sampled actions (logsumexp) | 10 |

Table 6: IQL hyperparameters.

| | Hyperparameter | Value |
|---|---|---|
| | Optimizer | Adam [18] |
| | Critic learning rate | 3e-4 |
| | Actor learning rate | 3e-4 |
| | Value learning rate | 3e-4 |
| | Mini-batch size | 256 |
| | Discount factor | 0.99 |
| | Target update rate | 5e-3 |
| | Learning rate decay | Cosine |
| IQL hyperparameters | Deterministic policy | True, Hopper Medium and Medium-replay |
| | | False, otherwise |
| | $\beta$ | 6.0, Hopper Medium-expert |
| | | 10.0, AntMaze |
| | | 3.0, otherwise |
| | $\tau$ | 0.9, AntMaze |
| | | 0.5, Hopper Medium-expert |
| | | 0.7, otherwise |
| | Critic hidden dim | 256 |
| | Critic hidden layers | 2 |
| | Critic activation function | ReLU |
| | Actor hidden dim | 256 |
| Architecture | Actor hidden layers | 2 |
| | Actor activation function | ReLU |
| | Value hidden dim | 256 |
| | Value hidden layers | 2 |
| | Value activation function | ReLU |

Table 7: AWAC hyperparameters.

| | Hyperparameter | Value |
|---|---|---|
| | Optimizer | Adam [18] |
| | Critic learning rate | 3e-4 |
| | Actor learning rate | 3e-4 |
| AWAC hyperparameters | Mini-batch size | 256 |
| | Discount factor | 0.99 |
| | Target update rate | 5e-3 |
| | $\lambda$ | 0.1, Maze2d, AntMaze |
| | | 0.3333, otherwise |
| | Critic hidden dim | 256 |
| | Critic hidden layers | 2 |
| | Critic activation function | ReLU |
| Architecture | Actor hidden dim | 256 |
| | Actor hidden layers | 2 |
| | Actor activation function | ReLU |

Table 8: SAC-$N$ and EDAC hyperparameters.

| | Hyperparameter | Value |
|---|---|---|
| SAC hyperparameters | Optimizer | Adam [18] |
| | Critic learning rate | 3e-4 |
| | Actor learning rate | 3e-4 |
| | $\alpha$ learning rate | 3e-4 |
| | Mini-batch size | 256 |
| | Discount factor | 0.99 |
| | Target update rate | 5e-3 |
| | Target entropy | -1 · Action Dim |
| Architecture | Critic hidden dim | 256 |
| | Critic hidden layers | 3 |
| | Critic activation function | ReLU |
| | Actor hidden dim | 256 |
| | Actor hidden layers | 3 |
| | Actor activation function | ReLU |
| SAC-N hyperparameters | Number of critics | 10, HalfCheetah |
| | | 20, Walker2d |
| | | 25, AntMaze |
| | | 200, Hopper Medium-expert, Medium-replay |
| | | 500, Hopper Medium |
| EDAC hyperparameters | Number of critics | 10, HalfCheetah |
| | | 10, Walker2d, AntMaze |
| | | 50, Hopper |
| | $\mu$ | 5.0, HalfCheetah Medium-expert, Walker2d Medium-expert |
| | | 1.0, otherwise |

Table 9: DT hyperparameters.

| | Hyperparameter | Value |
|---|---|---|
| DT hyperparameters | Optimizer | AdamW [27] |
| | Batch size | 256, AntMaze |
| | | 4096, otherwise |
| | Return-to-go conditioning | (12000, 6000), HalfCheetah |
| | | (3600, 1800), Hopper |
| | | (5000, 2500), Walker2d |
| | | (160, 80), Maze2d umaze |
| | | (280, 140), Maze2d medium and large |
| | | (1, 0.5), AntMaze |
| | Reward scale | 1.0, AntMaze |
| | | 0.001, otherwise |
| | Dropout | 0.1 |
| | Learning rate | 0.0008 |
| | Adam betas | (0.9, 0.999) |
| | Clip grad norm | 0.25 |
| | Weight decay | 0.0003 |
| | Total gradient steps | 100000 |
| | Linear warmup steps | 10000 |
| Architecture | Number of layers | 3 |
| | Number of attention heads | 1 |
| | Embedding dimension | 128 |
| | Activation function | GELU |

