# OpenReview forum: "CORL: Research-oriented Deep Offline Reinforcement Learning Library"
_NeurIPS.cc/2022/Workshop/Offline_RL — Offline RL Workshop NeurIPS 2022_

### Official Review · Reviewer_Xv7b · 2022-10-16

**Rating:** 8
**Confidence:** 5

**Review:**

## Summary
This paper proposes an open-source offline reinforcement learning codebase for the research research community. The codebase provides single-file, non-modular implementation for many offline RL algorithms, making it easy for researchers to prototype new algorithms.

## Review
Overall I believe this codebase could be a great contribution of the offline RL research community and therefore I strongly believe that this paper should be accepted.

### Pro
The codebase is really clean and provides a non-modular single-file implementation for many algorithms. This is especially important for research focused usage cases. Many offline RL codebases are deeply modular. While the modularity makes it easy to reuse code, such design also implies that the implementation of a single algorithm is usually distributed in many files, making it difficult to change the implementation for new algorithm prototype. Such codebases are often hard to use for algorithmic research where the alteration of algorithm happens rapidly. I believe that the author's codebase could fill in this gap and provides a good starting point for developing new algorithms.

The codebase implements many popular offline RL algorithms, making it more applicable for many different type of algorithm development and baseline evaluations.

### Con
The performance of certain algorithms still needs some improvement. For example, the performance of the CQL algorithm does not match that of another open source library I've used.